# Recent Progress in Development and Applications of Ionic Polymer–Metal Composite

**DOI:** 10.3390/mi13081290

**Published:** 2022-08-11

**Authors:** Si Won Park, Sang Jun Kim, Seong Hyun Park, Juyeon Lee, Hyungjun Kim, Min Ku Kim

**Affiliations:** 1Department of Mechanical Convergence Engineering, Hanyang University, Seoul 04763, Korea; 2School of Mechanical Engineering, Hanyang University, Seoul 04763, Korea; 3Department of Chemistry and Bioscience, Kumoh National Institute of Technology, 61 Daehak-ro, Gumi 39177, Korea

**Keywords:** IPMC, IPMC actuator, IPMC sensor, electroactive polymer

## Abstract

Electroactive polymer (EAP) is a polymer that reacts to electrical stimuli, such as voltage, and can be divided into electronic and ionic EAP by an electrical energy transfer mechanism within the polymer. The mechanism of ionic EAP is the movement of the positive ions inducing voltage change in the polymer membrane. Among the ionic EAPs, an ionic polymer–metal composite (IPMC) is composed of a metal electrode on the surface of the polymer membrane. A common material for the polymer membrane of IPMC is Nafion containing hydrogen ions, and platinum, gold, and silver are commonly used for the electrode. As a result, IPMC has advantages, such as low voltage requirements, large bending displacement, and bidirectional actuation. Manufacturing of IPMC is composed of preparing the polymer membrane and plating electrode. Preparation methods for the membrane include solution casting, hot pressing, and 3D printing. Meanwhile, electrode formation methods include electroless plating, electroplating, direct assembly process, and sputtering deposition. The manufactured IPMC is widely demonstrated in applications such as grippers, micro-pumps, biomedical, biomimetics, bending sensors, flow sensors, energy harvesters, biosensors, and humidity sensors. This paper will review the overall field of IPMC by demonstrating the categorization, principle, materials, and manufacturing method of IPMC and its applications.

## 1. Introduction

Polymer is a material **in which repeating subunits called** monomers **are** repeatedly bonded **to form a long chain**. Depending on the content, shape, distribution, and type of monomer, polymers have various physical, chemical, electrical, and thermal characteristics. Due to the various characteristics of the polymer, such as flexibility, being easily shaped, reversibility, chemical stability, and biocompatibility, research and applications of polymers are continuously growing in bio-engineering [1], robotics [2], batteries [3], sensors [4], 3-dimensional (3D) printing [5,6], and semiconductors [7]. Herein, external stimuli that can change the physical properties or shapes of the polymer include chemical [8], electrical [9], optical [10], and thermal stimuli [11]. These stimuli can be exploited to form the polymer into the desired shape and actuate to deform and produce force in specific situations. The electrical stimulus is most advantageous among the various stimuli because it can repeatedly return to its original structure after deformation without additional processing. Polymers that undergo any deformation due to electrical stimulation are called electroactive polymers (EAPs). The EAPs are activated by applying an electrical potential and deform into the intended shape and returning to the pre-deformation state when the potential is removed. As a result, EAPs are well suited for actuator and sensor applications where repeating deformation is required [12]. Activation with electricity provides a unique advantage for actuators or devices to be controlled simultaneously for a specific goal, such as a multi-functional actuation or multimodal sensing. Therefore, EAPs received much attention in applications where complex actuation or sensing is required, such as artificial muscles [13] or human interfacial sensors [14]. EAPs are similar to the piezoelectric effect [15], in which a mechanical deformation induces an electrical potential. EAPs can produce and store electricity, similar to a capacitor, in the event of mechanical deformation. Mechanical deformation can be measured from electrical signals (e.g., voltage, resistance). In general, EAPs respond to shape or property changes by producing electrical potential (e.g., voltage, current, electric field, etc.), and applied electrical charges cause mechanical deformation. These EAPs are largely categorized into ionic electroactive polymers (ionic EAPs) and electronic electroactive polymers (e-EAPs) [16,17]. Both ionic EAPs and e-EAPs are activated by applying electrical potential, but the main difference between the two is in the energy transfer process. While e-EAPs transfer energy by electric current, ionic EAPs transfer energy by ions. Since this energy transfer mechanisms work differently, each type has respective advantages and disadvantages, such as different operation voltage and varying actuation force. To overcome the disadvantages, various efforts and studies were conducted. Among ionic EAP types, polymer–metal composite (IPMC) is the focus of many studies attempting to overcome various disadvantages. This paper will first focus on understanding the field and principle of IPMC and examine its application based on the overall understanding.

## 2. Classification of EAP

EAP includes the type of polymers that react to electrical elements. When voltage or current is applied, EAPs react by mechanical deformation. In reverse, mechanical stimulation causes changes in electrical potential. EAPs are primarily divided into ionic EAP and e-EAP due to their differences in how the polymer membrane converts electrical energy into mechanical energy and vice versa. This energy transfer mechanism difference is the standard for the first classification, as shown in Figure 1. This section will briefly describe the working principle, subcategorization, advantages, and disadvantages of each EAP type.

### 2.1. Electronic Electroactive Polymer

This section will briefly describe the working principle and classification of e-EAP. The applied energy to the polymer of e-EAPs is transferred by an electric current through electrodes that cover a polymer membrane on either side. Then an electric field is formed within the polymer membrane of e-EAPs. As a result, Maxwell stress by the electric field within the membrane causes deformation of the shape of the polymer membrane [18,19]. *Figure 2a (upper) is the initial inactivated state of dielectric EAP and its composition. When activated with a potential, Maxwell stress is applied by the electric current, and the dielectric EAP deforms in thickness and area illustrated in Figure 2a (lower).* To form the electric field on the e-EAP device, various polymer materials are used, and the type of materials is a standard of the classification of e-EAP. With the standard, the e-EAPs can be classified into dielectric elastomer [20], ferroelectric polymer [21], electro-strictive material [22], and electro-viscoelastic elastomer [23]. The operation of e-EAPs through the electric current provides several advantages. It can operate with DC voltage and function well in dry conditions, enabling long-term operation without performance degradation. The e-EAP has a fast response time (in milliseconds) and its deformation can be maintained with constant DC voltage. In addition, the e-EAP devices exhibit relatively large operating forces and higher mechanical energy density compared to ionic EAPs. However, e-EAP has the disadvantage of requiring high operation voltage (~150 MV/m), requiring special circuits, and it can only deform in one direction [6,17,24]. As illustrated in Figure 2a, the changes in the area and thickness of the dielectric elastomer of e-EAPs occur when activated and result in actuation force. The dielectric elastomer can produce over 300% of strain, but it needs to strain before the fabrication of the dielectric e-EAP device [25]. A summary of the key advantages and disadvantages of e-EAP is provided in Table 1.

### 2.2. Ionic Electroactive Polymer

The working mechanism of ionic EAP occurs with the transfer of applied energy by ions. As the ions move from one crystal lattice to the other crystal lattice, the ion distribution becomes imbalanced within the membrane, resulting in a pressure gradient that leads to mechanical deformation. The initial inactive state of ionic EAP (Figure 2b upper) undergoes bending deformation (Figure 2b lower) by the pressure gradient within the polymer membrane of ionic EAP when activated by a potential. Activation voltages are applied to the polymer membrane by formed electrodes on the polymer membrane surfaces, as seen in Figure 2b. The polarization of a polymer from applied voltage results in the cations within the membrane moving toward the cathode, as well as electrons moving toward the anode. In this situation, cations are generated from the end of the polymer chain structure and are free to move within the structure, but the anions remain fixed in the chain structure [26]. Therefore, the ionic EAPs can be subdivided into different types according to the material composition of the polymer membrane containing the ions. Furthermore, the polymer membrane can be categorized into a conductive polymer, ionic polymer gel, electrical fluids, and ionic polymer–metal composites (Figure 1) [24]. Since the energy transfer of ionic EAPs occurs through the physical movement of ions, it has the disadvantage of having a relatively slow response time (range in seconds) and low operating force compared to e-EAPs. However, the advantages are that it can be operated at a relatively low voltage (<5 V), and it is capable of a wide range of bending displacements. Specifically, it has the benefit of being able to bend in opposite directions without the use of special design structures, as the bending direction depends on the polarity of the applied voltage. The key advantages and disadvantages of ionic EAP are summarized in Table 1. The rest of this review article will focus on IPMC, among ionic EAP types where the energy transfer occurs through the movement of ions within the polymer membrane. The main operation of IPMC is based on the transfer of electrical energy by the movement of ions. Since ions are required for the transfer of electrical energy, the polymer membrane must be ionized to introduce free ions, and a conductive electrode coating on the membrane is used to conduct electrical energy. As shown in Figure 3, at the initial state of IPMC (Figure 3a), there is a random mixture of free cations and electrons released from the ionized polymer. When a voltage is applied to the polymer membrane, the cations and electrons move towards the electrode with the opposite polarities. An illustration of this migration state of cations and electrons can be seen in Figure 3b. As a result, cations and electrons are concentrated along the polymer electrode interface, resulting in a pressure difference within the IPMC. The pressure difference causes the anode side to contract and the cathode side to expand, resulting in the entire IPMC sheet bending towards the contracted side (Figure 3c). However, once the applied voltage is turned off, the electrode polarity is removed, and the attracted cations and electrons return to a randomly mixed state. If the voltage is applied in reverse, cations and electrons once again move towards the opposite polarity electrode, causing the IPMC to bend in the opposite direction. Such a mechanism of IPMC allows deformation to occur in both directions, depending on the polarity of the applied voltage, which is one of the main advantages of IPMC over e-EAP [27,28,29].

## 3. Classification of Ionic EAP

The operation principle of EAPs is based on electrical stimulation. Among EAPs, ionic EAPs react through the movement of ions within the polymer membrane, transferring the electrical energy. Therefore, any type of polymer containing carrier ions can be used as the polymer membrane of an ionic EAP device. Polymers used in the polymer membrane of ionic EAPs can be largely divided into a conductive polymer, an ionic polymer gel, an electrochemical fluid, and an ionic polymer–metal composite according to their characteristics and material composition. Each type of ionic EAP is widely used in various applications, such as actuators [31,32,33], sensors [34], and energy storage capacitors [35]. Despite the advantages, limitations exist with ionic EAP, and various studies employed new types of materials to overcome them. For example, the metal electrode of IPMC was replaced with a conductive polymer to prevent the solvent evaporation of IPMC in a dry condition, which is one of the main limitations of IPMC. Many recent studies are currently conducted to combine IPMC with different ionic EAPs. The trend towards combining IPMC and other ionic EAP types is increasing and expected in future studies. The following section will describe the definition, material, advantages, and disadvantages of IPMC and other ionic EAP types.

### 3.1. Ionic Polymer–Metal Composite

The operation of any ionic EAP requires electrodes to form a polarity by an applied voltage for inducing a movement of positive ions within a polymer membrane. IPMC devices are constructed by applying such electrodes on the upper and lower surfaces of the ionized polymer membrane (Figure 3). IPMC maintains the advantages of ionic EAP, such as low voltage requirement, flexibility, bidirectional motion, and relatively fast actuation response time [36]. The strain of IPMC is similar to that of biological muscles and better than some actuators for artificial muscles, such as shape memory alloys, conductive polymers, and hydrogel actuators [37]. Recently, conductive polymers or polymers with nanoparticles (e.g., PEDOT, Nafion with carbon nanotube (CNT), etc.), with sufficient conductivity that can replace metallic electrodes on polymer membranes, were widely studied within the ionic polymer–polymer composite research [38,39]. For this reason, it is widely researched to further enhance the advantages of ionic EAP, while improving the disadvantages. Meanwhile, one of the disadvantages of IPMC is the low-frequency response range and performance instability under dry conditions. Recent studies [40,41] show improved device instability results by encapsulation with parylene. Other studies improved the performance of IPMC, such as the actuation speed and ionic conduction displacement rate, by replacing the polymer membrane solvent [42,43,44]. Figure 4a shows an example of IPMC bending actuation. In the study, IPMC, which is composed of a Nafion polymer membrane and Pt electrode strip device, was fixed on one end using a clamp, and voltage was applied for bending actuation. This experiment demonstrates the key advantage of IPMC: that it has a large bending displacement (>360°) and a fast reaction time of several seconds (<8 s) [30].

### 3.2. Other Types of Ionic EAP

A conductive polymer is an electrically conductive material and has a molecular structure in which electrons can move freely within the material. Examples of such conductive polymer include polyaniline (PANI), polypyrrole (PPy), polyacetylene (PAc), polythiophene (PTh), poly (p-phenylene), and poly (phenylene vinylene) [48,49,50]. Conductive polymers require low voltages for operation and can generate relatively large forces when actuated. In addition, they are commonly used in the biomedical field due to their biocompatibility [51]. However, deterioration due to fatigue from repeated movements is the limiting factor in some applications. Conductive polymers are generally formed by fused deposition methods, ink printing, casting, etc., and they are used in various applications, such as sensors [52], actuators [53] and flexible circuits [45]. Figure 4b is an example of a flexible circuit application using poly (3,4-ethylene dioxythiophene) polystyrene sulfonate (PEDOT: PSS) reported by Yuk et al., which is one of the widely used conductive polymers. PEDOT: PSS was used for printing a circuit that was encapsulated with Polydimethylsiloxane (PDMS). Another type of ionic EAP is ionic polymer gel. Polymers form an inner network, generally by hydrogen bonds, due to the chemical structure of their monomer [54,55]. With this network, a polymer can contain an ionic liquid and its phase becomes an elastic phase similar to rubber or gel, where ionic polymer gel (IPG) is a composite material in which an ionic liquid is fixed within a polymer matrix. When the surrounding pH level changes, IPG chemically reacts with $H^{+}$ ions and causes the volume of IPG to change. In addition, the changes can occur with sufficient levels of electrical stimulation. Examples of such IPG include poly (sodium acrylate), poly (vinyl alcohol), poly (vinylidene fluoride), hexafluoropropylene, polyacrylic acid, polyvalent alcohol (glycerol, polyvinyl alcohol), and polymethacrylic acid [56,57,58]. The advantage of IPG is that the energy density, the amount of electrical energy stored within a given volume in the case of IPG, is similar to the energy density of biological muscles while being biocompatible. IPG is also highly stretchable (>2000%) [55] and maintains conductivity at high temperatures (>175 °C) [59]. However, the disadvantages of IPG (device) are low ionic conductivities (<10 mS ${\mathrm{cm}}^{-1}$). IPG is being researched in applications such as displays [60], capacitors [61], actuators [46,62,63], and sensors [64]. For example, Palleau et al. [46] reported an actuator in 2013 where a tweezer was made with poly (sodium acrylate) hydrogel (Figure 4c). As shown in Figure 4c, the tweezer operated by applied voltage and grabbed the target object (0.1 g). The other type of ionic EAP is electrorheological (ER) fluid, which is an insulating fluid containing suspended particles that can be electrically activated. The physical properties of ionic polymer gel are closer to that of a solid, which requires a fabrication process before use, but electrorheological fluid does not need pre-forming to use because ER fluid normally behaves similarly to an ordinary fluid. However, under an electric field, the mixed particles in ER fluid react to the external electric field and align along the electric field. The aligned particles create a solid-like crystal lattice, and the creation of lattice structures causes changes in rigidity. Once the maximum rigidity is achieved, ER fluid can have a similar rigidity level to that of an elastomer. This phenomenon is not a reaction by a direct electrical current but by the alignment of suspended particles within an electric field. ER fluid has the disadvantage of requiring a relatively high voltage (>10 V). Examples of ER fluid include oxidized polyacrylonitrile, polyanilines, poly(p-phenylene), polypyrroles, and polythiophenes [65]. ER fluid is used in many applications, such as haptic actuators [47], mechanical dampers [66], and displays [67]. For example, Mazursky et al. [47] reported an application where a haptic actuator was constructed from ER fluid (Figure 4d). This haptic actuator was composed of contact space, which was in the middle of the device, and a non-contact space, which is the left and right sides of the device. Both spaces were covered with a PDMS membrane and filled with giant electrorheological (GER) fluids, as reported by Wen et al. [68]. As seen in Figure 4d, the contact space and passage had electrodes to which voltage applied. When voltage was applied, the particle in filled ER fluids would be aligned between the electrodes. As a result, the rigidity of ER fluids would change and the user would experience a resistive force.

## 4. Materials and Fabrication of IPMC

The materials and fabrication of IPMC can be explained in two sections because IPMC is composed of a polymer membrane and metal electrodes. The polymer membrane of IPMC serves as a storage and passage for ions that transfer electrical energy. In addition, the metal electrode plays the role of a passage through which electrical energy transmitted from the inside of the polymer by ions may flow. This structure of IPMC is a combination of an ionized polymer membrane and a metal electrode for better performance, such as bending displacement, actuation force, and fast response through the decrease in surface resistance. The polymer membrane materials will be explained using the most commonly used Nafion and other materials, and its fabrication methods will be demonstrated using commercial Nafion film as it is, solution casing, hot pressing, and 3D printing [69]. The electrode will deal with the formation of electrodes using not only metal electrodes but also polymer electrodes that were studied recently. Then, deposition methods of the electrodes, such as electroless plating, electroplating, direct assembly process, and solution deposition will be explained. Conventional IPMC is fabricated with commercial Nafion film and electroless plating. Other methods for the polymer membrane follow a similar overall manufacturing process to IPMC, but electroplating can be replaced with electroless plating at the electrode plating step.

### 4.1. Materials and Fabrication of Polymer Membrane

Nafion is a sulfonated tetrafluoroethylene-based fluoropolymer-copolymer, and it is the most used polymer membrane within IPMC. It is important for the activation and operation of IPMC that the ions move within the polymer membrane, and Nafion is well suited for this application because it is a synthetic polymer with ionomers from incorporated sulfonate groups. Other reasons for using Nafion are its chemical stability and endurance of fatigue by repeated operation [70]. In addition, the Nafion membrane can easily perform ion exchange only by being submerged in a solvent that is deionized water with ionic liquid (i.e., LiCl, EMI${\mathrm{BF}}_{4}$, and BMI${\mathrm{BF}}_{4}$) [71]. Nafion-based IPMCs are widely researched in applications such as grippers, bending actuators, sensors, micro-pumps, flow sensors, and biomimetic actuators. Despite the advantages, several limitations of Nafion still exist, such as short maintenance of the active state, deficient actuation force [72], critical solvent evaporation under dry conditions, the occurrence of electrolysis at operation voltage, and the high cost of manufacturing IPMC with Nafion [70]. Several studies employed alternate materials mixed with Nafion to overcome some of the issues. For example, Lee et al. [73] demonstrated an IPMC device of Nafion with polypyrrole (PPy) and alumina composite with a 2.7-times-higher bending performance compared with common Nafion IPMC. Meanwhile, Nguyen et al. [74] reported Nafion with layered silicate or fumed silica and achieved a three-times-larger blocking force than the conventional Nafion IPMC. Yip et al. [75] reported Nafion with multi-walled carbon nanotubes with two-times-larger force than only Nafion-based IPMC at 0.05 wt%. One of the methods used to enhance the properties of Nafion-based IPMC is altering the type of dopant used in the ion-exchange step, which is the primary charge carrier during its operation, resulting in the actuation of IPMC. Figure 5a illustrates the end chemical structure of Nafion made of $SO_{3}H^{-}$ [76]. The end structure of Nafion, SO_3_H^−^ is divided into SO_3_^2−^ and H^+^. Hydrogen ions perform the energy transfer shown in Figure 5b (undoped), where H^+^ ions are mainly used in the membrane. For improved performance of IPMC, H^+^ ions are exchanged with other ions through the doping method. This doping step can be performed at the preparing step or the final step during the fabrication of IPMC. Li^+^ [44], EMIBF_4_, and BMIBF_4_ [71], Na^+^, K^+^, Ca^+^, Mg^+^, Ba^2+^, TBA, and TMA [77] are possible ions for the doping process (i.e., ion-exchange). In general, Li^+^ ion demonstrates greater tip displacement. Figure 5b shows a schematic doping process, if H^+^ are exchanged to Li^+^. Although Nafion is widely used since its discovery in the late 1960s by Walther Grot of DuPont, several key disadvantages remain, such as being unable to maintain a continuous actuation state, low actuation force, and high manufacturing costs. *To make up for the shortages mentioned above, several additives, such as CNT, PPy/alumina, and Au-NP were mixed with Nafion, which resulted in enhanced performance, such as bending displacement, blocking force, energy efficiency, and ionic conductivity* [73,78,79,80]. Other polymers that can generate free ions within the membrane could be used as an alternate replacement for Nafion. Alternative polymers were studied as an alternative to Nafion for manufacturing IPMC. Materials such as polyvinylidene fluoride (PVDF) [70,72,81,82,83,84], sulfonated polyphenylene sulfone (sPPSU) [85,86], sulfonated poly(styrene-b-(ethylene-r-butadiene)-b-styrene (SSEBS) [87,88], polyvinyl alcohol (PVA) [87,89,90], and Kraton [91] are utilized for fabrication of IPMC. Many of the aforementioned polymers were used in addition to other additives during the membrane preparation.

After the material selection, fabrication of IPMC starts with forming the polymer membrane into the desired shape and thickness with selected materials. The solution casting method uses a polymer solution poured into a mold of the desired shape, as seen in Figure 6a. The casting and drying process can be repeated until the desired polymer thickness is formed. During the casting process, a porous structure can be formed on the surface by etching the silica precursor (tetraethyl orthosilicate in reference) mixed in a polymer membrane [92]. The solution casting method has the advantage of enabling a structure containing the multi-functional layers (i.e., actuation and sensing) because each layer was stacked successively on top of the previous layer. The hot pressing manufacturing method is where multiple prefabricated polymer films are stacked and pressed together with heat to obtain the desired thickness (Figure 6b). The multilayer IPMC device demonstrates improved bending and actuation force compared to the IPMC device made from single films of the same thickness. The membrane that underwent a hot pressing process contains a relatively greater number of ions compared to the commercially available Nafion film of the same thickness. The hot pressing method can be more cost-effective because the entire IPMC fabricating procedure (including the electrode) is accomplished in a single process. The 3D printing method shown in Figure 6c gained popularity in recent years and has the advantage of it being very simple to design the desired shape of the polymer membrane. In addition, the 3D printing method can create various surface profiles and roughness simultaneously [93]. Carrico et al. [94] and Yin et al. [95] demonstrated a 3D printing method to fabricate an IPMC actuator using a custom-made Nafion filament for the 3D printer.

### 4.2. Materials and Fabrication of Electrode

The basic operation of IPMC requires electrical stimulation or signals, and it is beneficial to improve the efficiency of electrical energy transfer through various means of electrode material selection and manufacturing methods. The operation performance of an IPMC, such as bending displacement, actuation force, and fast response can be improved by applying an electrode to a polymer membrane surface in order to increase its conductivity, rather than relying on the conductivity of the polymer. Figure 5c shows a schematic cross section of an electrode applied on the polymer membrane of an IPMC. During the manufacturing process of an IPMC, the metallic electrode penetrates the polymer membrane in the form of ions during the metal plating step of IPMC manufacturing. As illustrated in Figure 5c, the intermediate part between the electrode and the polymer membrane is formed. This intermediate region is formed by the roughening process when preparing the polymer membrane for a stronger bonding between the electrode and polymer membrane. Figure 5d illustrates the mechanism behind the stronger interface. Compared to the smooth raw surface, the roughened surface creates a larger bonding surface area, and it allows easier penetration of metal ions deeper into the polymer membrane. The roughening process during the preparation of a polymer membrane can be achieved through various methods, including grinding with sandpaper [40,96,97,98,99,100], the etching method [101,102,103,104,105], and creating a porous surface structure [92,106] at the initial stage of forming the polymer membrane. In general, the electrode materials are selected for their superior conductivity. Platinum (Pt), gold (Au), silver (Ag), and copper (Cu) are commonly used electrode materials for IPMC. The electrodes are formed on the polymer membrane surface either through electroplating or electroplating methods, which will be described later. The study that manufactured metal electrode is affected by thickness in the case of Au [91]. Recent studies utilized conductive polymers or a mixture of a conductive polymer–metal particles as an electrode instead of pure metals to overcome the disadvantages of metal electrodes, decreasing surface resistance and reducing manufacturing costs [107,108]. IPMC-utilizing polymer and polymer composites as electrodes are named ionic polymer–polymer composites (IPPC or IP^2^C), where PEDOT [38,109], graphene with EMIMBF_4_ film [110], sulfonated graphene oxide [111], CNT with Nafion [78], and silver nano-powder (Au-NP) with Nafion [79] are some of the examples. Although PEDOT: PSS can be used alone, it is often used as a mixture of the polymer and conductive nanoparticles. In a study by Rasouli et al. [80], the conductive polymer (PPy) mixed with the nanoparticles (carbon black, MWCNT) was used as an encapsulation coating on bulk Nafion membrane to ensure operational stability in dry conditions and further improve conductivity. IPPC demonstrated by Di Pasquale et al. only used PEDOT: PSS as electrodes, which was much cheaper to manufacture compared to IPMC with Pt electrode [108]. In future research, using a conductive polymer for an electrode can expect certain benefits of improved performance stability and lower manufacturing cost over bare metal electrode IPMC. Figure 7 shows the basic process of the electroless plating of IPMC, starting from the prepared polymer membrane [112]. The preparation of the polymer membrane consists of the production of polymer film and roughening of the film surface. Nafion is a commercially available polymer film, and it is commonly used as a polymer membrane. Some research demonstrated Nafion mixed with additives, such as CNT, PPy/alumina, and Au-NP [73,78,79,80]. Although Nafion is the most frequently used material in IPMC, other polymers, such as PVDF, sPPSU, SSEBS, and PVA were demonstrated as an alternative. After the material selection and formation of the polymer membrane, the fabrication of IPMC starts with the roughening process of the membrane surface by grinding with sandpaper [40,96,97,98,99,100] or etching [101,102,103,104,105]. The roughing process can sometimes be replaced by making the membrane porous at the membrane formation step [92]. Then the membrane is cleaned in purifying solution (i.e., HCl, H_2_O_2_). Next, the membrane is immersed in a plating solution (e.g., [Ag(NH_3_)_2_]OH, [Au(Phen)Cl_2_]Cl) that contains metal ions. During the plating process, the metal ions permeate into the polymer membrane and the roughened surface process ensures a more complete penetration of metal ions. The impregnated membrane is then soaked in a reduction solution (e.g., D-Glucose and NaSO_3_) to form the electrode with a chemical reaction. Lastly, the doping process (i.e., ion exchange) is performed, which converts the positive ions separated from the polymer membrane to other positive ions, which can generate more force inside the polymer. The ion exchange process improves the conductivity of IPMC and the response time. The general doping process was described in detail in Section 4.1. Electroplating is a method of forming electrodes on the polymer membrane of IPMC through electricity, while electroless plating is a chemical process of applying the electrodes. Manufacturing IPMC with the electroplating method follows the overall manufacturing process using electroless plating, such as in Figure 7, but is different in the impregnation step. As illustrated in Figure 6d, the electroplating method in the impregnation step requires a catalyst mesh, such as titanium mesh and the plating solution, which contains the metal ions for plating. The polymer membrane is connected to the cathode and the catalytic mesh to the anode. When an electric current flows, metal ions suspended in the plating solution are deposited on the surface of the polymer membrane by an oxidation–reduction reaction. After the plating step, the membrane is submerged in a reduction solution for the stabilization of metal ions deposited on the polymer membrane, which forms the IPMC electrode [113]. Other methods, such as the direct assembly process (DAP) can be employed to manufacture the electrode and the polymer membrane of IPMC. The direct assembly process method in Figure 6e is a technique for making electrodes with a mixture of conductive powder and ionomer solution, such as Nafion solution. The mixture is directly applied to an ionomer membrane (commonly Nafion) and hot-pressed to fix them. This technique results in more consistent and reliable performing IPMC devices [114,115]. The advantage of the DAP technique is that it does not require additional electrode forming steps because the electrode material is stacked with the polymer layer and hot-pressed together to form the IPMC. Another benefit of the DAP technique is that any type of ionomer, diluent, or conductive powder can be used, and the thickness and electrode configuration can be easily adjusted. Sputtering deposition is a method of spraying a solution of desired material to form a thin coating on any surface. This method can be used to produce both polymer film and metal electrodes of IPMC. The prepared solution proper for the purpose is spread out through the sputtering device, such as in Figure 6f. This method has the advantage that high melting point materials are easy to form electrodes by sputter, and the adhesion to the membrane is better than that of a common coating film. Trabia et al. [116] manufactured a custom polymer membrane shape with a shadow mask and an airbrush to spray the Nafion solution. In addition, Siripong et al. [117] sprayed gold electrodes on the Nafion film with 20–30 nm thick gold for sufficient adhesion between the metal and polymer membrane and Gudarzi et al. [118] sputtered the IPMC with acrylic acid to form a waterproofing barrier.

## 5. Application of IPMC

The operation principle of IPMC was explained in the previous section. Applied electrical potential causes the positive ions in the polymer membrane to move towards the electrode interface, resulting in a pressure gradient caused by osmotic pressure from the imbalance of the ions. Here, the electrical energy is converted to mechanical energy, which appears as mechanical deformation. However, this process can occur in reverse. IPMC deformation, by an external mechanical stimulation, creates an imbalance of ions in the polymer membrane, resulting in the electrical potential difference of its electrodes [119]. The direction of the energy conversion determines the function of IPMC as an actuator or a sensor. In the following sections, the application of IPMC as an actuator was classified into soft robotic grippers, micro-pumps, and biomimetic actuators. In addition, the IPMC sensor application was classified into bending sensors, fluid flow sensors, energy harvesting devices, wearable biosensors, and humidity sensors by utilizing specific characteristics of IPMC.

### 5.1. Actuator

IPMC is based on an ionized polymer membrane with a conductive electrode coating. The ions inside the polymer membrane responsible for electrical energy are usually the sulfonated polymer and H^+^ ions. When an electrical potential is applied, polarity is imparted to each electrode and the ions move towards the opposite polarity. The ions concentrated along the interface of the polymer membrane combined with the electrode create a difference in the density of the ions on either side of the neutral mechanical plane within the membrane, which causes an osmotic pressure phenomenon. The pressure gradient within the IPMC causes expansion on one side and contraction on the other, resulting in bending movement [38]. IPMC actuators can utilize such bending motions in the various actuation applications listed in the following sections.

#### 5.1.1. Gripper

IPMC has the advantage of being able to achieve large bending displacement with a small voltage and being able to manufacture an actuator with a desired scale and shape. Moreover, IPMC can achieve high flexibility due to its polymer membrane, which is well suited as an actuator for a soft robotic gripper. An IPMC gripper is produced with the goal of picking things up with large bending. Figure 8a demonstrates an IPMC gripper with a wire connected to activate the gripper. When the IPMC gripper is well approached to target the object, the IPMC operates to pick up the object by a bending movement caused by applied voltage. Different types of grippers can be made depending on the number of IPMC actuators used. Ford et al. [120] reported that single and double IPMC grippers are made with Nafion membrane and Pt electrode. Explaining with Figure 8a if an IPMC gripper is a set of an actuating part of IPMC actuator and an assist part made of steel or something similar, that gripper is a one-finger gripper [121,122]. In the case of a two-finger gripper, it is a set of both actuating parts of IPMC [123,124,125,126,127]. As a result, in [120], it was reported that the one-finger gripper had better performance. Chen et al. [122] suggested that the assist part of the IPMC gripper set is replaced with a sensor to perform a special role, calibration between voltage, and operating force. As a result, an object on the millimeter scale can be picked up more completely. Jain et al. [126] and Inamuddin et al. [127] suggested an IPMC gripper consisted of a two-finger gripper and an additional IPMC actuator as a joint. This IPMC gripper with a joint was for not only picking, but also an additional movement of the gripper itself, as seen in Figure 8a (upper left). Meanwhile, similar to Figure 8a (lower left), the IPMC gripper was equipped with a supporting part capable of performing gripping action, which attempted to lift objects on a smaller scale [101,128,129].

#### 5.1.2. Micro-Pump

The key advantage of IPMC is that it is capable of bidirectional deformation, depending on the applied voltage polarity [36]. Nguyen et al. [130,131] well demonstrated this property of IPMC by fabricating a diaphragm pump. Figure 8b is an example of an IPMC micro-pump, which consists of an inlet and outlet channel with an IPMC diaphragm membrane. When the IPMC diaphragm is activated in the upward direction, increasing internal volume and decreasing pressure, the liquid flows through the inlet channel. When the polarity of the voltage is changed, the IPMC diaphragm is deformed in the downward direction, and the liquid is pushed through the outlet channel. The overall micro-pump device and exposed IPMC diaphragm can be seen in the inset image of Figure 8b. The right side of Figure 8b is an exploded illustration of components and a construction of the micro-pump. The flap valve layer prevents the backward flow from the outlet valve and the inlet valve during the pump operation. In this application, the polymer membrane was created via the solution casting method using a mixture of Nafion and modified silica solution. The electrode was similarly fabricated with the solution mixed with Nafion, layered silicate, and silver nano-powder [132,133].

#### 5.1.3. Biomedical

IPMC coated with biocompatible elastomer demonstrates flexible and reversible properties, making them well suited for applications in biomedical devices. Feng et al. [134] reported a tactile bump array actuator in 2018. This actuator was composed of IPMC actuator arrays with a PDMS bump placed on the tip of IPMC actuators. When a voltage is applied, the target bump is pushed through the printed circuit board (PCB) by the IPMC actuator and pressed against the finger in contact with the device providing tactile feedback. The tactile bump array was Nafion film, cleaned in an HCl solution, and immersed in Pt solution. Afterwards, prefabricated copper tape, using photolithography, was attached to the Nafion membrane. The membrane with the copper tape would undergo the reduction step, while the copper tape was removed from the membrane. The placement of copper tape during the ion exchange process formed the patterned electrode on the IPMC. As shown in Figure 8c (right), the actuator was completed by assembling the PCB, the PDMS bump, the patterned electrode IPMC, PDMS backing, and a titanium support plate in order. An external voltage source operated the assembled actuator device (~2 mm thick) to transmit the actuation force to the fingertip through the PDMS bump on the IPMC (Figure 8c, left). There was a study on applying IPMC actuator to an optical lens. Horiuchi et al. [135] reported an IPMC actuator integrated within an intraocular lens (IOL) system (Figure 8d). The IPMC actuator was placed within the IOL, manipulating the thickness of the lens, thereby controlling the amount of light transmission. The change in thickness of the lens can be adjusted by sets of IPMC actuator wings pushing the soft material in both directions. Experimental results demonstrate that the system could accommodate up to an amplitude of the spherical equivalent of 0.8 D with a voltage of ±1.3 V.

#### 5.1.4. Biomimetic Application

Biomimicking was a source of inspiration for many generations of scientists. The biomimetic application focuses on studying a more efficient mechanical movement of living things in nature. IPMC is widely used in biomimetic applications as an artificial muscle due to its bidirectional movement, large actuation force, and flexibility. Ma et al. [30] reported the fast-flapping movement of a dragonfly wing with an IPMC actuator. Figure 9a is an image of the dragonfly robot with wings made from IPMC. The dragonfly robot was operated with an AC voltage of 19 Hz at 5 V, shown in Figure 9a (right). The IPMC was manufactured with a Nafion polymer membrane with an electroless plating method to form a Pt electrode. The IPMC underwent an ion exchange process in LiCl, as ions increase electrical energy transfer. Similarly, Zhao et al. [100] reported a robot imitating the movement of a beetle. This robot had a flexible wing, which was driven by IPMC artificial muscles. The flexible wings could flap driven by 4.5 V at 0.5 Hz with a maximum displacement of 6.4 mm. Another example of IPMC in biomimicry was replicating the locomotion of various forms of marine life, imitating a marine animal’s back and forth tail movement [124,136,137,138,139,140,141,142,143,144]. Among many demonstrations of robots that can swim in water with an IPMC as a tail, the dolphin robot reported by Shen et al. [137] achieved the highest driving velocity of 46 mm/s with a thrust efficiency of 66.5%. A similar example of an IMPC manta ray robot capable of underwater locomotion, reported by Chen et al. [145], is shown in Figure 9b. The construction of the manta ray robot was composed of an IPMC wing for primary locomotion and a plastic tail for stabilizing the body. The plastic body housed the control circuit and battery. The IPMC wing of the manta ray robot was fabricated from a Nafion membrane with a Pt electrode and PDMS encapsulation for water protection. The robot was driven in the water with 3.3 V at 0.4 Hz and achieved a swimming speed of 0.42 cm/s. Other studies demonstrated applications of IPMC replicating the locomotion of turtles [146], jellyfish [99,147], and lobster [148]. Figure 9c is a robot that imitates the movements of a caterpillar reported by Carrico et al. [149]. The body of the caterpillar robot was 3D printed with Nafion precursor filaments and Pt electrodes were formed with electroless plating. The robot consists of four operating parts, divided into two working portions, forming a body, and each of the two remaining operating parts serves as a leg. Each part of the body was extended and contracted in sequence along the direction of movement. At the same time, the legs would open and close systematically to provide a fixture to push against the platform to move forward.

### 5.2. Sensor

The working principle of an IPMC is based on the electrical energy transfer by ions in the polymer membrane of the IPMC. This electrical energy applied to an IPMC induces mechanical deformation. However, this process can occur in inverse. When an external force is applied to an IPMC, structure deformation causes the change in ion concentration, resulting in a potential difference between the electrodes. Almost all IPMC sensors are based on this principle. This section will review various IPMC-based sensors, such as a bending sensor, displacement sensor, fluid flow sensor, energy harvester, biosensor, and humidity sensor.

#### 5.2.1. Bending and Displacement Sensor

The general form factor of a bending or displacement sensor is shaped similar to a cantilever in order to allow external forces to easily bend the sensor in one direction [40,41,150,151,152,153,154,155,156]. Figure 10a (left) is a demonstration of the calibration setup of the IPMC bending sensor. One side of the IPMC bending sensor was fixed and the other side was free to move. A mechanical force producer and load cell are installed on the free side of the beam to produce the mechanical deformation to the IPMC bending sensor. The laser sensor detects IPMC sensor displacement correlating to applied force by the load cell and the electrical signal generated by the IPMC. Figure 10a (right) shows that the output voltage changes according to the deformation of the IPMC cantilever sensor [157]. Lei et al. [40] and Song et al. [41] demonstrated improved sensing resolution by applying parylene coating to the IPMC sensor. In [40], the IPMC sensor was manufactured with commercially sourced Nafion film roughened with sandpaper and Pt electrode by electroless plating. Parylene was coated on the surface of the IPMC sensor. The IPMC with coated parylene was immersed in water to retain enough moisture for better sensitivity. In [41], Nafion film was treated with a hot pressing method, and Pt electrode was produced through electroless plating. Lastly, ion exchange was performed, replacing the lithium ion and finishing with the parylene coating. As a result, the reported IPMC sensor achieved a sensor resolution of 0.06 mV and a sensitivity of 12 mV/cm.

#### 5.2.2. Flow Sensor

Another use of the IPMC bending sensor can be seen in Figure 10b. Stalbaum et al. [159] reported a flat IPMC flow sensor resembling a bending sensor. The flow sensor was placed in a fluid flow path and deformed according to the flow rate. The amount of deformation produced a correlating voltage potential, enabling an estimation of the flow rate measurement. Furthermore, Abdulsadda et al. [160] and Hong et al. [102] demonstrated a study of sensing the fluid flow rate by placing an array of bending sensors in the fluid path with an estimated velocity error of 0.1 cm/s and a vibration amplitude error of 0.19 mm (Figure 10b, upper left). Tsugawa et al. [161] suggested a slender IPMC flow sensor for further flow sensing. The suggested flow sensors had four electrodes placed around a Nafion polymer membrane. The electrodes on the Nafion membrane reacted to another direction of the flow by making a pair with the opposite electrode for the voltage charge due to the ion imbalance of the polymer membrane. A common IPMC flow sensor, which had two electrodes, could detect the forward direction of flow as well as the backward direction of it. However, four electrode IPMC flow sensors could detect flow rate in forward, backward, left, and right directions. The IPMC used in this study was manufactured in two different forms, one in hollow tube form and the other in solid rod form. Attempting to apply the hollow Nafion tube for the polymer membrane would be expected, and merits flowing chemicals or fluids, forwarding power and signal wires, or applying tools, such as cameras. The polymer membrane was prepared from a commercially available Nafion tube, and the rod type was made by a cutting a Nafion film sheet. Pt electrode was deposited by electroless plating. After the electrode deposition, the manufactured IPMC sensor underwent the milling process for removing the Pt electrode on an IPMC. The purpose of the milling Pt electrode of the IPMC was the division of deposited electrodes for forming four electrodes.

#### 5.2.3. Energy Harvesting

Based on a similar mechanism of sensing application, IPMC can also be used in energy harvesting [162]. Giacomello et al. [158] reported an example of a flexible flag with IPMC installed on its surface and placed under water for energy harvesting (Figure 10b). The water flow causes the flag to flutter, which in turn causes the IPMC to deform, generating energy by voltage charge. Harvested power was reported around ~10^−10^ W at a water flow rate of 0.6–1.1 m/s. Cellini et al. [163] also reported energy with a similar type of energy harvesting device. The rotating turbine connected to a system of gears and a crank translated the rotating motion to an oscillating linear motion, which was connected to an IPMC strip. The repeated bending of the IPMC generated an electrical potential, and the external circuitry harvested the energy. As a result, harvested power ranged from 1 pW to 1 nW from a flow rate from 0.23 to 0.54 m/s. A different study by Cha et al. [96,164] presented an energy harvesting system where IPMC was attached to an artificial fish robot [164] and a thin beam [96] for energy harvesting. The IPMCs used in the mentioned studies were created through an electroless plating process using commercially sourced Nafion film. Tiwari et al. [165] reported a disc-type IPMC energy harvesting device. The key benefit of a disc-type form factor is that it can generate energy from deformations in all three directions. The disc-type IPMC was fixed with a clamp and a mechanical shaker deformed disc-generating energy. The disc-type IPMC was created by hot pressing Nafion pellets into a mold, and a conductive silver paint was used to form the electrode. The disc IPMC was demonstrated by charging a NiMH battery actuated by the shaker system.

#### 5.2.4. Biosensor

IPMC can generate currents even with a small local mechanical stimulus. In addition, IPMC is flexible, which makes it ideal for a wearable mechanical sensor. In 1999, Keshavarzi et al. [166] reported a study to measure blood pressure using IPMC. The sensor was fabricated with Nafion film and a Pt electrode formed with an electroless plating method. The reported sensing range was 15–165 mmHg calibrated with a load cell. The measured pressure range was beyond the normal pressure range of 80–120 mmHg, demonstrating a sufficient measurement range. Subsequently, in 2018, Chattaraj et al. reported a wearable IPMC device capable of measuring the blood pulse rate from the wrist. This Nafion/Pt electrode IPMC strip was integrated into a standard stethoscope. The pulse from the wrist was simultaneously measured with both the stethoscope and IPMC strip. IPMC strip was validated on multiple subjects and demonstrated an error range of 4–15% compared to a standard plethysmography device. Multiple studies reported devices to detect motion changes by attaching an IPMC sensor to the human body. Liu et al. [167] reported an IPMC sensor modified with a graphene electrode that achieved a response speed of ~50 ms and 1.3 mV output voltage under 1.8% strain induced by natural body motion. The IPMC sensor was formed with a Nafion membrane and Au electrode by electroless plating. Moreover, an electrolyte of the IPMC sensor was changed from water to 1-ethyl-3-methylimidazolium bis (trifluoromethyl sulfonyl) imide (EMITFSI) to achieve higher sensitivity. The graphene was further applied by solution casting followed by a hot pressing method. This Graphene–Au electrode IPMC sensor was capable of measuring large deformation motions of the body, such as wrist movement and posture change, with sufficient resolution to measure small changes, such as pulse and touch. In 2018, Ming et al. [103] reported a smart glove system integrated with multiple IPMC sensors placed at the fingertips and the joints of the index and middle finger. The IPMC was fabricated with a roughened Nafion membrane and electroless plated with gold. The authors reported a more uniform electroless deposition by separating and aligning each Nafion sample with a fluid bed-like reactor, overcoming the disadvantages of conventional electroless plating, such as long processing time and batch instability. The IPMC-integrated smart glove was used to measure blood pulse, recognition of braille language, and robot hand control (Figure 10c). Lee et al. [97] reported an IPMC-based throat sensor that detects mechanical movement from the throat. The sensor was composed of a Nafion membrane formed by hot pressing against sandpaper, creating a porous surface structure and a gold electrode by electroless plating. The resulting IPMC sensor was attached to the throat area (Figure 10d) to detect throat movement during breathing, talking, and volumetric changes in the throat. By measuring volumetric changes and processing the data with an artificial intelligence algorithm, the IPMC sensor was able to automatically detect coughing, humming, nodding, and swallowing at an accuracy of up to 95%.

#### 5.2.5. Humidity Sensor

The actuation of IPMC results from the concentration difference of ions, which leads to osmotic pressure, causing physical deformation. However, the osmotic pressure inside the polymer membrane results from the movement of water molecules. The evaporation phenomenon of solvent, which is dependent on the humidity of the environment, affects the water content, ultimately altering its actuation level. This enables IPMC to be used as a humidity sensor. According to a study conducted by Shoji et al. [168], an IPMC actuator placed in 30% relative humidity (RH) and a 90% RH environment demonstrated a 10-fold displacement difference. Other studies were conducted to investigate the correlation between the humidity (water uptake ratio) and performance in voltage output, bending displacement, and ionic diffusivity of the IPMC membrane [107,169,170]. Esmaeli et al. [98] proposed an IPMC device that can be used simultaneously as an actuator and a humidity sensor by utilizing the correlation between IPMC capacitance and humidity of the surrounding environment. The polymer membrane surface was modified with oxygen plasma treatment, followed by the electron beam deposition of titanium and gold electrode to increase electrode adhesion strength and improve sensor sensitivity. The resulting device demonstrated an RH sensing range between 40% and 90%, with the fastest response and a sensitivity value of 13.43%. In another study, Beigi et al. [171] created a Nafion doped with Co-Al-layered double hydroxide (LDH) with Pt electrodes. The fastest response to humidity change was achieved with a 1% Co-Al LDH level. The resulting IPMC demonstrated the highest sensitivity of ~408 millivolts per humidity at a humidity range from 20 to 90%.

## 6. Conclusions

Polymers are studied in various fields, taking advantage of their chemical, thermal, electrical, and physical properties. Although there are some differences in physical properties depending on the material, polymers, in general, are easy to process, flexible, and reversible. Polymers that have electrical characteristics, such as conductivity, dielectric, and are ion-containing, are especially useful in industry and applications affecting human life. Therefore, research on electrically activated polymers are continuously studied with great interest. EAPs are widely used in the field of soft robotics, sensors, micro-pumps, biomedical, display, optical devices, biomimetic robots, and batteries. EAPs can be divided into an ionic EAP and an electronic EAP by the electrical energy transfer mechanism. An ionic EAP transfers the electrical energy through the movement of ions within the polymer part of ionic EAP. Due to this energy transfer mechanism, ionic EAP has the advantage of low activation voltage and large bending displacement compared to electronic EAP. However, the relatively slow response time and the low actuation force of ionic EAP were the main impediments. Some of the recent approach was to create a metallic electrode covering the polymer membrane of ionic EAP, which resulted in an IPMC. It is classified as an intermediate stage between organic and inorganic, incorporating strengths from both categories. Many polymer materials containing ions for the energy transfer within an IPMC device are used for the polymer membrane of IPMC. Among many polymers for IPMC, Nafion is the most common material for the membrane because of it contains the hydrogen ions, which play the role of energy transfer. The metallic electrode of the IPMC device is made up of common electrode materials, such as silver, gold, and platinum. Recently, the polymer electrode was studied to replace the metallic electrode for improvement in the IPMC performance under the name of IPPC. Solution casting, hot pressing, and 3D printing are the manufacturing methods of the polymer membrane. Electroless plating and electroplating are used for the adhesion of metals to the polymer membrane. On the one hand, a direct assembly process and sputtering deposition can be used for the manufacturing of both the polymer membrane and metal electrode. The current trend is heading towards replacing the metallic electrode with an organic conductive polymer. As IPMC is further improved, its direction is expected to proceed towards construction entirely of polymers (IPPC), further enhancing its biocompatibility, actuation, and sensing performance. IPMC is already being applied in soft robotics (gripper and biomimetic) and biomedical fields (micro-pumps, haptic or optical devices, and biosensors), and it is expected that it will be able to assist bodily function through artificial muscles or smart biomedical devices. Its functions can be enhanced with the addition of AI data processing. Therefore, understanding IPMC will be helpful for future studies of electrical polymers. For that purpose, the definition of IPMC, comparison of IPMC with other ionic EAP types, the principle of IPMC, the material and manufacturing method of IPMC, and applications of IPMC are described in this review.

## Figures and Tables

**Figure 1 micromachines-13-01290-f001:**
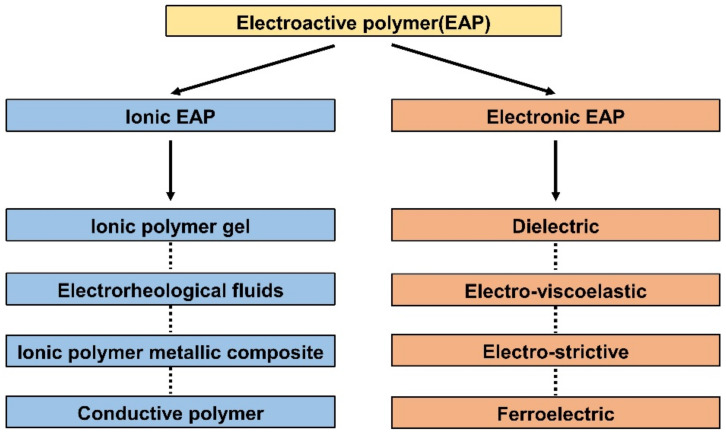
Classification of various electroactive polymers. Adapted from Rahman et al. [16] with permission from MDPI, Copyright (2021).

**Figure 2 micromachines-13-01290-f002:**
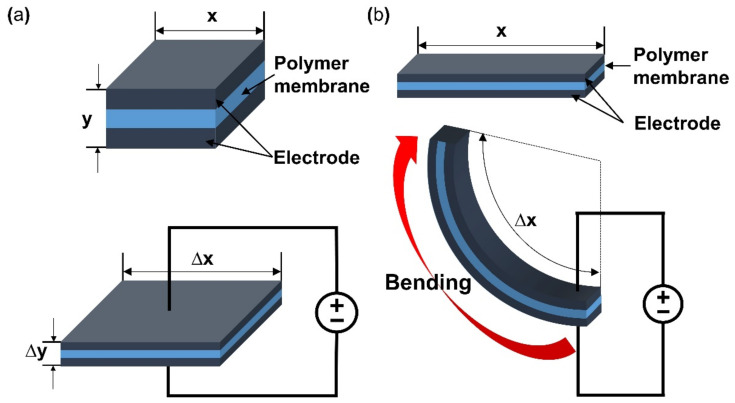
A Visualization of deformation during activation of (**a**) dielectric EAP and (**b**) ionic EAP. Adapted from Engel et al. [6].

**Figure 3 micromachines-13-01290-f003:**
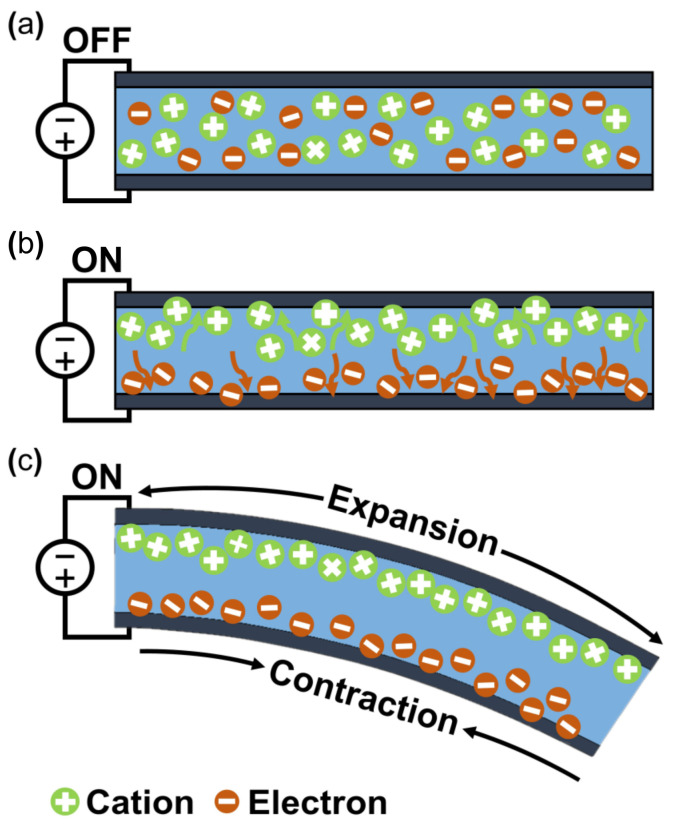
Actuation principle of IPMC. (**a**) Initial state. (**b**) Ion migration state. (**c**) Bend state. Adapted from Ma et al. [30] with permission from Wiley-VCH Verlag GmbH, Copyright (2019).

**Figure 4 micromachines-13-01290-f004:**
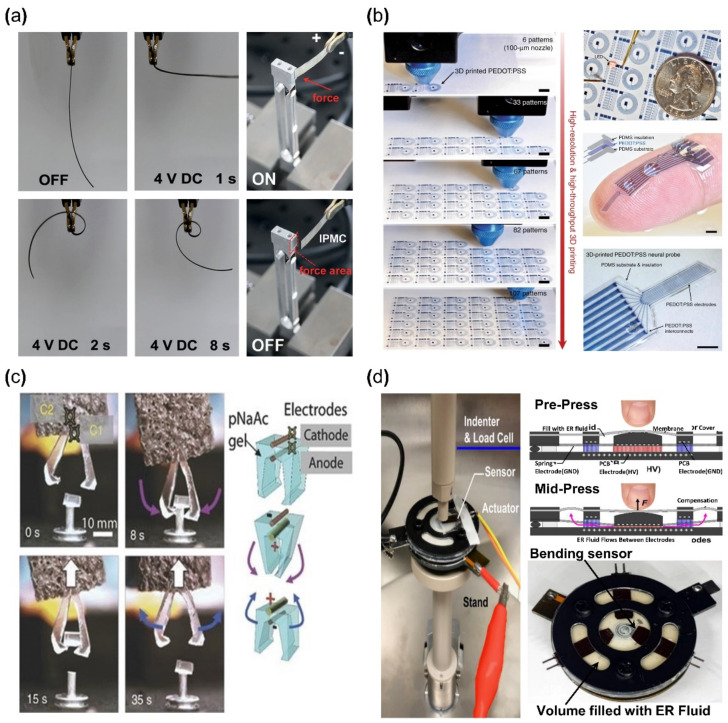
Application examples of ionic EAP devices. (**a**) Bending actuator; bending aspect of IPMC (left) and force measurement method of IPMC (right). Adapted from Ma et al. [30] with permission from Wiley-VCH Verlag GmbH, Copyright (2019). (**b**) Demonstration of basic circuit 3D printed with PEDOT: PSS. Adapted from Yuk et al. [45] with permission from Nature Portfolio, Copyright (2020). (**c**) Mechanical tweezer constructed from ionic gel. Adapted from Palleau et al. [46] with permission from Nature Portfolio., Copyright (2013). (**d**) Haptic actuator module made from electrorheological fluids. Adapted from Mazursky et al. [47] with permission from MDPI, Copyright (2021).

**Figure 5 micromachines-13-01290-f005:**
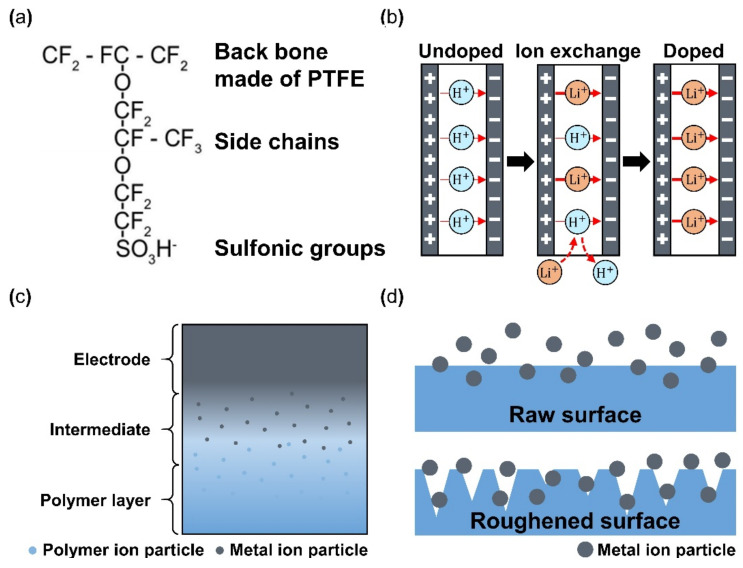
(**a**) Chemical structure of Nafion. Adapted from Valenzuela et al. [76] with permission from Trans Tech Publications LTD, Copyright (2009). (**b**) Schematic of ion exchange process during polymer doping. (**c**) Cross section view of an electrode plated polymer membrane. Adapted from Ma et al. [30]. (**d**) Metal ion particle penetration comparison between an unprocessed smooth surface and the mechanically/chemically processed rough surface of polymer membrane.

**Figure 6 micromachines-13-01290-f006:**
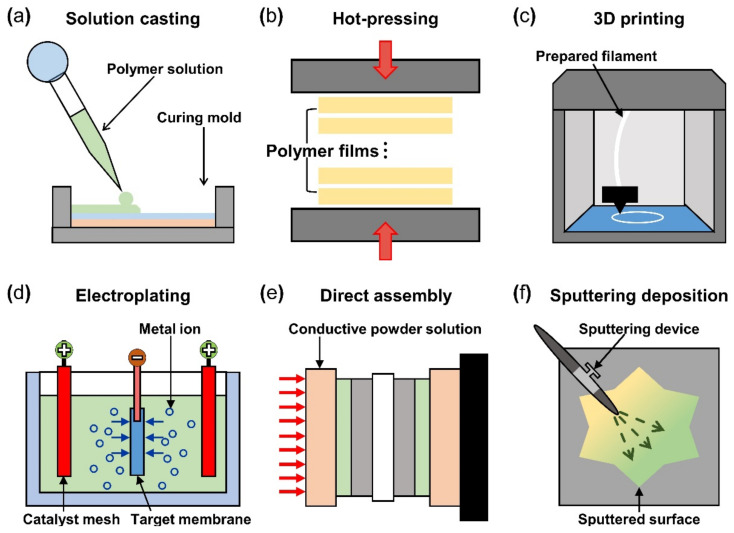
Schematics of IPMC polymer membrane manufacturing method, (**a**) solution casting method, (**b**) hot pressing method, and (**c**) 3D printing method. IPMC electrode manufacturing method (**d**) electro plating method, (**e**) direct assembly process method, and (**f**) sputtering deposition method. Adapted from Yang et al. [69].

**Figure 7 micromachines-13-01290-f007:**
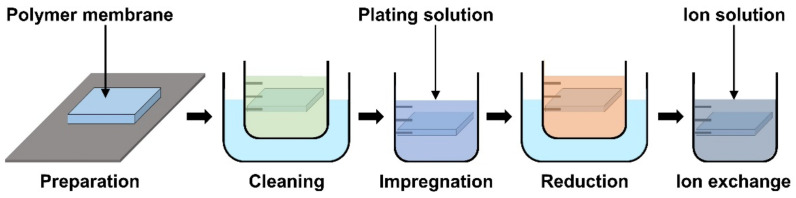
General manufacturing steps of IPMC using electroless plating. Adapted from Yang et al. [112].

**Figure 8 micromachines-13-01290-f008:**
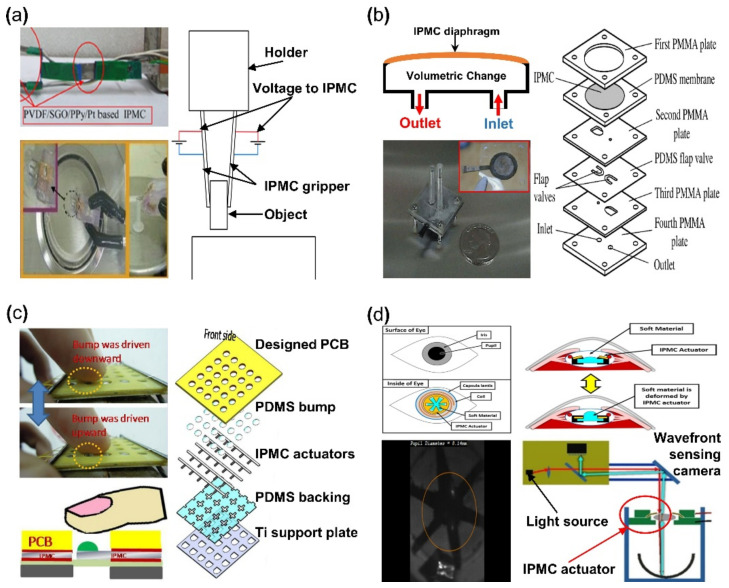
Applications of IPMC actuator. (**a**) Demonstration of IPMC gripper (left). Adapted from Inamuddin et al. [127] with permission from Springer Nature, Copyright (2019) and Feng et al. [101] with permission from Elsevier, Copyright (2007), and corresponding experiment setup (right). (**b**) Demonstration of IPMC micro-pump (left) and exploded schematic of IPMC micro-pump construction (right). Adapted from Nguyen et al. [131] with permission from Elsevier, Copyright (2007). (**c**) Demonstration of IPMC tactile device actuation (left) and exploded schematic tactile device construction. Adapted from Feng et al. [134] with permission from Elsevier, Copyright (2018). (**d**) Example of IPMC intraocular lens and working principle (upper) and experimental demonstration of IPMC IOL (lower). Adapted from Horiuchi et al. [135] with permission from Optical Soc Amer, Copyright (2016).

**Figure 9 micromachines-13-01290-f009:**
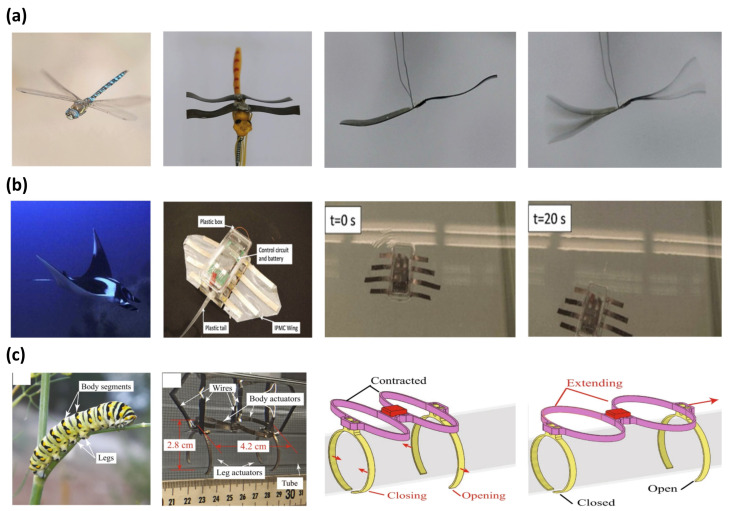
Demonstration of biomimetic IPMC actuator inspired by (**a**) dragonfly, adapted from Ma et al. [30] with permission from Wiley-VCH Verlag GmbH, Copyright (2019). (**b**) Manta ray, adapted from Chen et al. [145] with permission from Elsevier, Copyright (2011), and (**c**) Caterpillar. Adapted from Carrico et al. [149] with permission from Nature Portfolio, Copyright (2019).

**Figure 10 micromachines-13-01290-f010:**
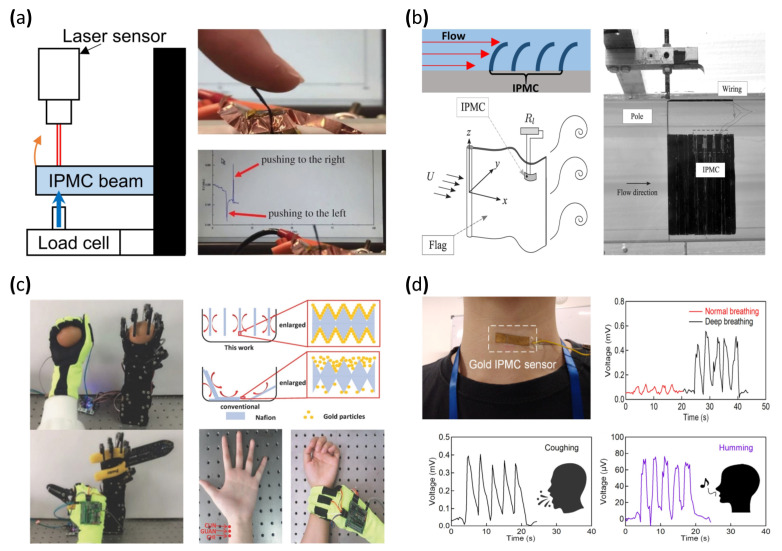
Demonstration of IPMC based sensor. (**a**) Experiment setup of an IPMC bending sensor (left) and the corresponding signal response. Adapted from Wang et al. [157] with permission from The Royal Society, Copyright (2016). (**b**) Demonstration of IPMC fluid flow sensor/energy harvester. Adapted from Giacomello et al. [158] with permission from AIP Publishing, Copyright (2011). (**c**) Demonstration of a fluid bed-like reactor (right) and photograph of blood diagnosis and robot arm control. Adapted from Ming et al. [103] with permission from Wiley, Copyright (2018). (**d**) Photograph of an IPMC throat sensor and corresponding voltage signal graphs. Adapted from Lee et al. [97] with permission from MDPI, Copyright (2021).

**Table 1 micromachines-13-01290-t001:** Characteristics comparison of ionic and electronic EAP. Adapted from Engel et al. [6] and Bar-Cohen et al. [17].

EAPs Type	Advantages	Disadvantages
Electronic	Long actuation time in room conditions.	High voltage requirement (20~150 MV/m).
EAP	Fast actuation response time (msec).	Unidirectional operation due to electrostriction effect.
	Large actuation force.	Requiring pre-strain at >300%.
	High energy density (mechanical).	
	Maintain deformation under DC voltage.	
Ionic EAP	Extensive bending (on average).	Unstainable strain under DC.
	Low voltage actuation.	Slow response time range in seconds.
	Bidirectional operation with voltage polarity.	Low actuation force in bending.
		Electrolyte requires humid condition.
		Electrolysis occurs at >1.23 V when a system involves water.

## Data Availability

Not applicable.

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
