# Peer review of "Recent Progress in Development and Applications of Ionic Polymer–Metal Composite"

_micromachines, 2022, doi:10.3390/mi13081290_

Round 1

Reviewer 1 Report

The authors give a comprehensive overview of the development and application progress of ionic polymer metal composite. The manuscript is fully discussed, written in a standard, and can be published.

Reviewer 2 Report

1. Language of the abstract needs to be refined. 

2. It is suggested to merge the fifth part into the fourth part.

3. The conclusion is suggested to be elaborated in sections, which is more clear.

4.It is suggested that the author consider simplifying the part of the article except IPMC to highlight the focus of the article.

Reviewer 3 Report

The review paper on "development and applications of ionic polymer metal composite" is well organized and will helpful to the materials community. Few below points to be checked before acceptance:

1) Figur 2 : each of the sub-figures should be addressed.

2) the abbreviation PCB should be written in full form during its first mention in the text.

3) Sequence of the figure should be maintained. e.g: Figure 2(b) addressed prior to figure 2(a)

Reviewer 4 Report

1.      A grammatical error in Section 4.2 heading

2.      Justify the statement: “The use of a conductive polymer for an electrode can have certain advantages in terms of improved performance and economic value over bare metal electrode IPMC.

3.      Which is more efficient electroless plating or electroplating? If so, why?

4.      Elaborate this statement with suitable discussion “To make up for such shortages, several additives such as dopants were mixed with Nafion.”
